# The VincerEmo Pilot Study: Prospective Analysis of Controlled Physical Activity in People with severe Hemophilia

**DOI:** 10.3390/jcm14186652

**Published:** 2025-09-21

**Authors:** Federica Valeri, Cristina Dainese, Piera Merli, Mariella Galizia, Samuel Agostino, Nicolas Cunsolo, Carola Sella, Alessandra Valpreda, Mariagiulia Bailon, Marco Miniotti, Annamaria Porreca, Giuseppe Massazza, Benedetto Bruno, Alessandra Borchiellini

**Affiliations:** 1Regional Centre for Hemorrhagic and Thrombotic Diseases, AOU Città della Salute e della Scienza, 10126 Turin, Italy; fvaleri@cittadellasalute.to.it (F.V.); aborchiellini@cittadellasalute.to.it (A.B.); 2Division of Hematology, AOU Città della Salute e della Scienza, University of Turin, 10124 Turin, Italy; sella.carola@gmail.com (C.S.); benedetto.bruno@unito.it (B.B.); 3Department of Orthopedy and Traumatology and Riabilitation, AOU Città della Salute e della Scienza, 10126 Turin, Italy; pieramerli@libero.it (P.M.); giuseppe.massazza@unito.it (G.M.); 4Sisport Spa SSD, 10151 Turin, Italy; mariella.galizia@fcagroup.com (M.G.); samuel.agostino@unito.it (S.A.); nicolas.cunsolo@edu.unito.it (N.C.); 5Department of Molecular Biotechnology and Health Sciences, University of Turin, 10124 Turin, Italy; 6Clinical Biochemistry Laboratory, AOU Città della Salute e della Scienza, 10126 Turin, Italy; avalpreda@cittadellasalute.to.it; 7Clinical Psychology Unit, Department of Neuroscience, University of Turin, 10124 Turin, Italy; giulia.bailon@hotmail.com (M.B.); marco.miniotti@unito.it (M.M.); 8Department for the Promotion of Human Science and Quality of Life, San Raffaele University, 00166 Rome, Italy; annamaria.porreca@uniroma5.it

**Keywords:** hemophilia, physical activity, joint health, prophylaxis

## Abstract

**Background/Objectives**: The approach to physical activity in people with hemophilia (PwH) is still conditioned by many difficulties. Thus, a prospective observational pilot study has been carried out aiming to evaluate how an adequate and controlled training program can slow down the onset or evolution of arthropathy and improve musculoskeletal health and quality of life. **Methods**: Performed from April 2022 to April 2023, this study involved nine severe hemophilic A and B patients, aged > 18 years old, on regular prophylaxis with replacement products. Participants, without changing the usual prophylaxis schedule and maintaining a trough level of at least 20% FVIII/FIX before training, were involved in physical activity twice a week. **Results**: After 12 months, no increase in annual bleeding ratio (ABR) was observed, and baseline joint status (as assessable by HEAD US score, HJHS, and NRS) was maintained. Even if not statistically significant, a trend toward improvement in mean HEAD US score (15.55 vs. 13.11) and HJHS (14.4 vs. 11) from baseline was observed. Some of the physical tests performed showed a significant improvement at 6 months and 12 months from baseline (5 Rep Sit to Stand, Sit and Reach, and 6-minute Walking Test), meaning an improvement in leg strength, dorsal flexibility, and aerobic resistance. **Conclusions**: This is the first pilot study evaluating at 360 degrees the safety and impact of a controlled physical activity in PwH. No participant experienced bleedings or a worsening in joint status, but they experienced an improvement in articular functionality. Without changing the usual prophylaxis, scheduling training sessions according to individual pharmacokinetics turned out to be a safe and a cost-effective approach.

## 1. Introduction

Hemophilia is an inherited hematological bleeding disorder caused by a deficiency in clotting factor VIII (FVIII) or IX (FIX). The musculoskeletal system is frequently affected by repeated joint bleeds in people with hemophilia (PWH). Intra-articular hemorrhages lead to irreversible changes in the cartilage and bone tissue, resulting in motor impairment, muscle hypotrophy from non-use, postural instability, and pain. These conditions make PWH more vulnerable to physical stress and predisposed to easy bleeding [1,2,3]. For these reasons, the approach to sports in PWH is conditioned by many difficulties; due to physical limitations and to prevent traumatic bleeding, these patients have been discouraged from practicing sports. However, nowadays, available therapies for hemophilia treatment have led to a change in attitude towards physical activity, though patients must, in any case, be carefully monitored by hematologists and physiatrists [4].

Physical activity is part of a healthy lifestyle, promotes general well-being, and is recommended by the World Health Organization (WHO). In addition, regular physical exercise should be recommended for PWH to increase muscle strength and proprioception, maintain joint mobility and bone health, improve physical functioning, reduce the impact of bone ageing, and potentially reduce the bleeding risk [5,6]. In particular, physical activity, especially swimming, is recommended for hemophiliacs to improve their quality of life and physical condition, increase strength and resistance, and reduce the risk of musculoskeletal lesions [6,7,8]. A recent randomized study demonstrated how progressive strength training with elastic resistance performed twice a week for eight weeks is safe and effective in people with hemophilia. It was shown to improve muscle strength and functional capacity, reduce general pain, as well as improve self-rated health status and desire to exercise [9]. Also, aerobic exercise training significantly improves pulmonary functions in hemophilia A patients, making it a potential addition to medical treatment [10]. Physical activity and exercise are also associated with great health benefits, such as a lower risk of cardiovascular disease, hypertension, obesity, diabetes, depression, and osteoporosis. Their potential to aid in the treatment and management of these comorbidities is becoming more pertinent in the context of chronic health, particularly in the ageing population with hemophilia. The choice of target factor levels is a critical component of the correct management of sport in PWH. It is challenging to establish the true relationship between specific physical activity and the exact prophylaxis levels required to reduce even a transient increase in bleeding risk without modifying the treatment regimen. No specific information on safe trough levels is available; due to a lack of scientific evidence from large, prospective controlled clinical trials regarding physical activity in PWH, no evidence-based guidelines regarding participation in exercise and sport are available [11].

In a systematic review published by Iorio et al., an expert panel proposed target plasma levels of factor VIII/IX to be adopted for the optimal prevention of bleeding in hemophilia, also in the context of moderate-to-high-risk physical activity, but these data are to be confirmed in clinical practice [12].

## 2. Materials and Methods

This prospective, single-center pilot study was conducted at the Regional Reference Center for Thrombotic and Hemorrhagic Disorders of Adults of the AOU City of Health and Science of Turin between April 2022 and April 2023. This study received approval from the Local Ethics Committee in October 2019 (protocol number 0098190). Included subjects signed informed consent. Data were collected at the baseline screening visit, at a six-month interim visit, and at 12 months (end-of-study visit).

Inclusion criteria considered were FVIII or FIX < 2%, aged 18–60 years, with a regular unmodified prophylaxis schedule for at least six months before enrolment, ABR ≤ 2, and articular function ≥ 50% of the normal range to be eligible. Patient characteristics are shown in Table 1.

The primary endpoint of this pilot study was to explore in real life the safety of a 20% FVIII/FIX trough level when performing physical activity in terms of annual bleeding ratio (ABR). Secondary endpoints consist of the evaluation of osteoarticular and muscular health of PWH assessed by Hemophilic Joint Health Status (HJHS) score [13] and HEAD US score through US evaluation [14]. Indeed, we collect information about pain using the NRS (Numeric pain Rating Scale). Concerning physical activities, the impact of training on cardiovascular and musculoskeletal systems was evaluated using the following tests: tapping test (Tap), Hand Grip strength (HGS); 5 rep Sit to Stand (RTS); Sit and Reach (SaR); Flamingo Balance Test (FB); and 6 min walking test (SMWT). The following assessments were carried out at the screening visit (T0), at the first time point (6 months, T1), and at the second time point (12 months, T2):Screening visit (T0): Hematological evaluation with inclusion and exclusion criteria verification, confirmation of anti-hemorrhagic prophylaxis schedule and compliance verification, ABR and pain according to NRS (Numeric pain Rating Scale) evaluation, individual pharmacokinetic curve analysis and definition of weekly training days, and articular echographic evaluation according to the HEAD-US score protocol. Physiatric evaluation for joint health with HJHS, articular range, muscular strength, and postural assessment. Physical baseline tests: performed at the training center with dedicated personal trainers6-month follow-up (T1): Hematological evaluation with inclusion and exclusion criteria revision, antihemorrhagic prophylaxis schedule and compliance verification, ABR and pain according to NRS evaluation, and articular US evaluation according to the HEAD-US score protocol. Physiatric evaluation for joint health with HJHS articular range, muscular strength, and postural evaluation. Physical interim tests.12-month follow-up (T2): Hematological evaluation with ABR and pain according to NRS evaluation, and articular US evaluation according to the HEAD-US score protocol. Physiatric evaluation for joint health with HJHS and articular range, muscular strength, and postural evaluation. Physical final tests.

Furthermore, a psychologist using the focus group research method with study participants and operators conducted a psychological evaluation.

FVIII/FIX trough levels maintained to perform training sessions were ≥20%, based on the statement reported in the Delphi consensus. For each patient, a calendar with infusion days and allowed training days was calculated, based on single pharmacokinetic curves.

Physical activity and training programs were conducted at the Sisport^®^ Training Centre of Turin, under the supervision of trainers specifically educated in hemophilic patient management and criticality. The intervention follows the recommendations outlined in the Negreir et al. study [11]. Due to the absence of randomized clinical trials, formal evidence-based guidelines have not been established.

The participating subjects were divided equally into 2 groups (group A, group B).

Each group participated in 2 training sessions per week lasting an hour each for 12 months. One weekly training session was devoted to joint mobility activities, core strengthening, along with exercises dedicated to balance. Exercises dedicated to joint mobility included exercises focused on each body joint, with controlled and cyclic movements that would allow the improvement of both mobility and vascularization and warming up of the joints, which are dedicated to muscle strengthening. During this training session, particular attention was paid to strengthening the abdominal wall and lumbar area, working on posture; the result is to maintain balance in static form.

The second training session alternated weekly between muscle-strengthening activities and water aerobics activities in the pool. Regarding muscle-strengthening activities, muscle reconditioning work was implemented, aiming to exercise all the major muscle groups, both upper and lower body. Subsequently, exercises dedicated to both strength and muscular endurance of the main muscle groups were conducted through mainly isometric (low joint impact) but also isotonic work, with exercises initially free-body and later using light overloads, elastic bands, weight balls, and suspension training cables.

As for pool training, mainly water aerobic exercises were carried out, with aerobic workouts taking advantage of the buoyancy that water exerts on the body. The initial part included a warm-up, followed by walking/running in place, squats, jumping jacks, upper- and lower-limb contraction, stretching exercises, and finishing with relaxation exercises. Exercises were conducted with both free-body and stability-enhancing equipment such as a pull buoy, noodle, and pull kick. Overall, the type of activity suggested is considered high-impact, adapted to individual participants’ baseline statuses, target joints, and the improvements noted throughout the study period.

From the physical ability point of view, participants were evaluated at a baseline, T1, and T2 for hand strength (Handgrip Test), neuro-motor ability (Finger Tapping Test), lower-limb strength (5 Times Sit To Stand Test), static balance (Flamingo Balance Test), muscle-tendon flexibility (Sit and Reach Test), and cardio-respiratory function (6 Minutes Walking Test).

Patients were instructed to immediately report to Hemophilia Centre hematologists in case of trauma, injury, bleeding, or any other possible issue connected to the hemorrhagic disorder or physical activity.

Descriptive statistics were expressed as median and q1  = first quartile, q3  = third quartile, for continuous variables, and absolute frequency (*n*) and column percentage (%) for categorical. A normal distribution was verified with the Shapiro–Wilk test, and, for all variables, the null hypothesis of normality was not verified, with a significance level of 95%. The Friedman test was used to assess differences over time for continuous variables. When the non-parametric analysis of variance resulted in a significant difference between time points, Dunn’s post hoc test, with a Bonferroni correction, was used to compute multiple pairwise comparisons. All statistical tests were 2-sided, with a significance level set at *p*  <  0.05. Analyses were performed using the R software environment for statistical computing and graphics (version 4.1; http://www.r-project.org/).

## 3. Results

Eight patients with severe hemophilia A and one with severe hemophilia B fulfilled the inclusion criteria and were enrolled in this study. All participants were able to complete the 12-month evaluation. The median age of the patients was 37.00 years (33.00–44.00), and the median trough level during prophylaxis was 2.80% (1.50–5.40) with various treatment regimens. The median number of previous target joints was 2.00 (2.00–2.00). Three patients were treated with standard half-life (SHL) products, and six patients received extended half-life (EHL) products. Baseline ABR was 0 for eight patients, while only one patient had a baseline ABR of 2. At both T1 and T2, the ABR became 0 for all patients. Nearly 60% of sports sessions were performed with a basal FVIII of 20%, and the remaining 40% with an FVIII level > 20%. The results are reported in Table 2 and Table 3.

No bleeding events or other sport-related adverse events occurred, and consequently, no participants had to withdraw from this study. Only one participant experienced a progressive exacerbation of right knee pain without evidence of a recent hemorrhage. Following a psychiatric assessment, this symptomatology was ascribed to a flare-up of a prior inflammatory issue. The patient continued physical activity and achieved a resolution of the painful condition through the administration of anti-inflammatory agents (COX inhibitors).

## 4. Discussion

Despite the limitation of enrolling a very small number of participants, this is the first study to prospectively explore regular physical activity in PWH during standard prophylaxis. This study was performed with a minimum trough factor level of 20% before sports sessions, without any modification to dose or frequency. Furthermore, the sample is homogeneous, with patients sharing similar articular characteristics. It is important to note, as a crucial aspect of the study design and its innovative message, that the prophylaxis schedule for each participant was not modified at the beginning or during this study. In other words, each participant continued their usual prophylaxis schedule, and physical activity was distributed throughout the week according to PK curves. This allowed sports to be performed on days when the trough level of circulating factor was at least 20% before the training session. This approach, based on individual PK curves, ensures that the patient’s prophylaxis does not need to be adapted to their sports schedule. Instead, sports activity can seamlessly fit into the patient’s daily life without requiring changes to their habits. Furthermore, this approach is also pharmacoeconomically sustainable, given the intensity of the proposed physical activity. As mentioned before, 60% of training sessions were performed with an FVIII level of 20%, while the remaining sessions were conducted with an FVIII level > 20%, in accordance with individual PK curves. Our main result is that an FVIII/FIX level of 20% proved to be safe for the study participants and for the type of activity proposed, as no bleeding episodes were registered during physical activity. The fact that an infusion is not strictly necessary before every session allows for greater flexibility in choosing training days throughout the week, according to the regular infusion schedule. This increases patient involvement by making sports sessions easier to plan in daily life and, consequently, increases the rate of participation, with evident benefits for general well-being.

We observed two phases of patient involvement across the entire year: in the first six months, the participation rate was high (80% of sessions were attended), with a significant decrease in the last six months (with only four patients attending more than 60% of training sessions). This latter rate overlapped with the general population’s involvement in recreational sports at the Sisport Sports Center during the same period. The study results, investigated across three distinct time points—baseline, 6 months, and 12-month follow-up—clearly reflect this participation trend. We observed some improvements in the examined variables at 6 months that were subsequently lost at 12 months. Regarding articular function, physical exercise can lead to increased production of synovial fluid and a resulting increase in nutrient diffusion to the cartilage. This process reduces joint degeneration and prevents major osteo-myo-articular impairment. Furthermore, resistance training improves the strength of the periarticular muscles, reduces bone loss, and promotes cartilage lubrication, thereby reducing stiffness and pain [15].

In this study, we observed that six out of nine patients (66%) showed an improvement in their HEAD US score at the six-month evaluation, specifically due to a reduction in pre-existing synovitis. This data may reflect a decrease in microbleeds that, even if not clinically apparent, can lead to a slow deterioration of joint structure. For the remaining three patients (34%), no differences were detected at the six-month US evaluation; these patients already had a very good baseline HEAD US score, indicating good joint health without synovitis and with only mild cartilage or bone disease. We did not observe this improving trend in the second part of this study (from T1 to T2), likely due to the decreased frequency of sports sessions. Specifically, three patients (34%) showed a worsening in their T2 US evaluation compared to T1, while in the other six patients (66%), the HEAD US score did not change between T1 and T2.

Regarding the HJHS score, five patients had the same score at T1 as at T0. In three patients, we observed an improvement in the HJHS score at T1 compared with T0, mainly due to decreased pain and improved global gait. One patient’s score worsened because of a mild loss of right ankle extension. At the T2 timepoint, six patients’ HJHS scores did not change compared with T1, two patients’ scores ameliorated (indicating better pain control and stiffness), and one patient’s score worsened.

Regarding the evaluation of physical abilities, a statistically significant improvement across the 12 months was observed in leg muscle strength, as evaluated by the 5-Times Sit-to-Stand Test. The scientific literature claims that good levels of muscle strength reduce the chance of experiencing various functional limitations by 50%, which normal physiological decline may bring [16]. Clinical trials performed in arthritis patients [17,18] have shown that strength training improves physical performance, disability, and pain. For people with hemophilia whose muscles are very weak or whose joints are quite painful with movement, isometric exercises—which involve muscle contraction without joint movement—are an excellent way to begin strength training. Strengthening weak muscles, particularly the quadriceps, may stabilize joints affected by hemophilic arthropathy. Because strength, muscle mass, and gait are closely intertwined, resistance training can also improve walking ability in adults with articular damage [19]. Although not statistically significant, an improving trend across 12 months was also observed for balance, evidenced by improvements in the bilateral Flamingo Balance Test, for muscle-tendon flexibility with the Sit-and-Reach test, and for cardiorespiratory function with the 6-Minute Walking Test. It is known that good balance corresponds to good postural control, which greatly implies better movement ability. However, balance training can be challenging for PWH with severe arthropathy, particularly those with multiple joint involvements. Stretching exercises, such as those performed in this study and whose benefits were evaluated with the Flamingo Balance Test and the Sit-and-Reach Test, can improve the flexibility of muscles and tendons around arthritic joints, easing pain and improving balance. Lastly, improving cardio-respiratory function can lead to better blood pressure control and weight maintenance, reducing the risk of obesity, cardiovascular complications, and type 2 diabetes. No significant changes across the time points were observed for the Tapping and Handgrip tests.

## 5. Limitations of the Study

This study presents several limitations related to its design and exploratory nature. First, the absence of a control group does not allow us to fully exclude the possibility that the observed changes in clinical and functional parameters (e.g., HEAD-US, HJHS, physical performance tests) may be due to spontaneous fluctuations over time or to external factors not directly related to the intervention, such as potential seasonal variations in joint health.

Moreover, the small sample size, although homogeneous in terms of clinical characteristics and treatment regimen, limits the generalizability of the findings and prevents definitive conclusions regarding the effectiveness of physical activity in this patient population. The pilot design prioritized the assessment of the safety and feasibility of a personalized approach based on individual pharmacokinetic profiles, without modifying the ongoing prophylaxis regimen. While this allowed for the collection of preliminary data in a realistic and controlled context, it does not allow for a clear isolation of the intervention effect from potential confounding variables.

Finally, adherence to the training program was high during the first six months but declined in the second half of this study. This decrease likely influenced the trend of results over time, with an initial improvement not consistently maintained at the 12-month follow-up.

Given these limitations, future studies with randomized controlled designs, larger sample sizes, and longer durations will be necessary to more robustly assess the efficacy of physical activity in the management of hemophilia, including in patients with varying degrees of arthropathy.

## 6. Conclusions

Aware of the very small number of patients included, our study is the first to show how standard prophylaxis, without any additional infusions, could allow for safe and beneficial physical activity to be performed with a minimum trough level of 20%. We would also like to emphasize that this approach is cost-effective, since no additional costs for extra infusions are required. Further interventional studies with a larger number of patients and a broader range of arthropathy are needed to deeply explore the potential benefits of safe physical activity in hemophilia patients.

## Figures and Tables

**Table 1 jcm-14-06652-t001:** Patient characteristics.

	Type of Hemophilia	Weight (kg)	Height (cm)	BMI	FVIII/FIX Product	Prophylaxis Regimen	Trough Level (%)	No. of Target Joints
**P1**	A	80	185	23.4	Octocog alfa	37 UI/kg every 72 h	5	2
**P2**	A	74	170	25.6	Octocog alfa	40 UI/kg every 72 h	2	2
**P3**	A	90	178	28.4	Plasma-derived Factor VIII	20 UI/kg every 72 h	1	2
**P4**	A	70	178	2.1	Damoctocog alfa pegol	42 UI/kg every 96 h	6	2
**P5**	A	75	176	24.2	Efmoroctocog alfa	40 UI/kg every 72 h	1	2
**P6**	A	56	170	19.4	Damoctocog alfa pegol	35 UI/kg every 96 h	3	3
**P7**	A	71	182	21.4	Efmoroctocog alfa	60 UI/kg every 72 h	3	4
**P8**	A	65	165	23.9	Damoctocog alfa pegol	46 UI/kg every 120 h	1	1
**P9**	B	77	175	25.1	Albutrepenonacog alfa	50 UI/kg every 240 h	12	1

**Table 2 jcm-14-06652-t002:** Hematological results. Descriptive statistics expressed as median (q1  =  first; q3  = third) quartile of the considered parameters at screening, 6 months follow-up and 12 months follow-up (end of the study). ABR: annual bleeding ratio. NRS: Numeric pain Rating Scale for pain assessment, HJHS: Haemophilia Joint Health Score, HEAD US score: Haemophilia Early Arthropathy Detection with Ultrasound.

Variables	T0: Screening	T1: 6-Month Follow-Up	T2: 12-Month Follow-Up	*p*-Value
**ABR**	0.00 (0.00; 2.00)	0.00 (0.00; 0.00)	0.00 (0.00–0.00)	0.524
**NRS**	3.00 (0.00; 7.00)	3.00 (3.00; 4.00)	3.00 (2.00; 4.00)	0.971
**HJHS**	14.5 (5.50; 27.5)	8.00 (6.00; 21.00)	14.5 (5.50; 25.5)	0.985
**HEAD US**	9.00 (7.00; 23.0)	9.00 (6.00; 16.00)	14.0 (6.75; 16.2)	0.524

**Table 3 jcm-14-06652-t003:** Physical results. Descriptive statistics expressed as median (q1  =  first; q3  = third) quartile of the considered parameters at screening and 6-month and 12-month follow-ups. Tap: tapping test (taps/second); HGS: Hand Grip strength (kg); RTS: 5 rep Sit to Stand (seconds); SaR: Sit and Reach (centimeters); FB: Flamingo Balance Test (falls/60 s); Six: 6 min walking test (meters); R: right; L: left. We used the bold to highlight the statistically significant results.

Variables	T0: Screening	T1: 6-Month Follow-Up	T2: 12-Month Follow-Up	*p*-Value
**Tap R**	60.0 (58.0; 63.0)	61.0 (60.2; 61.0)	60.5 (58.8; 61.2)	0.798
**Tap L**	59.0 (58.0; 62.0)	61.0 (58.0; 61.8)	61.5 (60.8; 62.2)	0.568
**HGS R**	44.0 (37.7; 45.3)	41.0 (38.4; 43.6)	41.6 (39.4; 45.3)	0.936
**HGS L**	38.1 (31.0; 44.7)	38.1 (35.5; 44.6)	38.3 (36.0; 43.0)	0.874
**RTS**	11.1 (10.0; 12.9)	8.66 (8.51; 9.62)	8.64 (7.04; 9.07)	**0.002**
**SaR**	26.5 (24.0; 34.0)	36.8 (36.5; 37.8)	35.2 (32.0; 36.2)	**0.027**
**FB R**	11.0 (9.00; 16.0)	12.0 (8.00; 14.0)	8.50 (7.25; 12.0)	0.605
**FB L**	8.00 (7.00; 9.00)	9.00 (5.75; 11.5)	7.00 (5.75; 8.50)	0.771
**Six**	595 (493; 607)	668 (544; 760)	628 (595; 662)	0.146

## Data Availability

The data presented in this study are available on request from the corresponding author (data are not publicly available due to privacy restrictions).

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
