# Peer review of "The VincerEmo Pilot Study: Prospective Analysis of Controlled Physical Activity in People with severe Hemophilia"

_jcm, 2025, doi:10.3390/jcm14186652_

Round 1

Reviewer 1 Report

Comments and Suggestions for Authors

General Comments:

  1. Relevance and Novelty: The study addresses an important and timely topic, the safety and efficacy of structured physical activity in people with severe hemophilia under regular prophylaxis. The concept of personalizing physical activity scheduling using individual pharmacokinetics is innovative and clinically relevant.
  2. Study Design and Cohort: The prospective design and structured follow-up over 12 months add strength to the study.
  3. Statistical Analysis: The statistical methods are appropriate for a small pilot cohort.

Specific Comments:

Methods:

  • Clarify if any participants dropped out or had adverse events during training.
  • The use of individual pharmacokinetic curves is commendable and should be highlighted more clearly as a key strength.

Discussion:

  • Well-written with good contextualization in existing literature.
  • The link between patient adherence and maintenance of benefit is an important point; consider offering suggestions for improving adherence over time.
  • The suggestion that physical activity at trough levels of ≥20% is safe is promising but needs cautious language due to the small cohort.

Conclusions:

  • The term cost-effective should be used carefully; no formal cost analysis was provided.

Minor Points:

  • Proofread the manuscript for typographical errors.

Final Recommendation:

This is a valuable pilot study with clinically relevant findings. The manuscript would benefit from minor revisions to improve clarity, readability, and cautious interpretation of data due to the sample size. The authors are encouraged to elaborate on the feasibility and next steps for larger-scale studies.

Comments on the Quality of English Language

Quality of English language is good but need some proofreading for minor typo errors.

Author Response

Dear Reviewer

Thank you for your constructive comments and observations. My responses to your points are detailed below.

  1. Clarify if any participants dropped out or had adverse events during training. The use of individual pharmacokinetic curves is commendable and should be highlighted more clearly as a key strength. As requested, we added the information on page 5, lines 194-200. We have emphasized the approach based on the use of PK curves on page 7, lines 224-228.
  2. The link between patient adherence and maintenance of benefit is an important point; consider offering suggestions for improving adherence over time. The suggestion that physical activity at trough levels of ≥20% is safe is promising but needs cautious language due to the small cohort. Regarding strategies for improving adherence to physical activity, the data is comparable to that of the general population starting a physical activity program, with higher adherence in the first months followed by a gradual decline. Therefore, we did not provide suggestions for improving a parameter that is generalizable to the wider population and would require more extensive reflections beyond the scope of this study. We have emphasized that FVIII/FIX levels 20% proved to be safe in the study population. Given the small sample size, we intentionally avoided providing an absolute recommendation (page 7, lines 230-232).
  3. The term cost-effective should be used carefully; no formal cost analysis was provided. The reviewer's observation is correct; formal pharmacoeconomic studies were not conducted in our research. However, our use of the term 'cost-effective' was intended to highlight that no additional factor was consumed compared to the baseline. This is because the participants' standard prophylaxis remained unchanged from before the training program, and no extra infusions were needed for intercurrent bleeding events, thus keeping pharmacological expenditure constan
  4. Proofread the manuscript for typographical errors. We have made the requested changes 

Reviewer 2 Report

Comments and Suggestions for Authors

The work is very interesting; in fact, I had already read the Poster Abstract published in the journal Haemophilia (February 2024) and was eager to learn more about this study. That said, there are some aspects that I believe need to be modified or improved:

  1. In my opinion, the MATERIALS and METHODS section is insufficient. You only mention one hour of exercise, but you do not specify the type of exercise performed, which joints were involved, or whether the exercise was high- or low-impact… and, obviously, you also do not provide any dosage of the exercise. You state that you followed the recommendations of Negrier et al. (2013), but these recommendations are very general (low-impact activities, safe exercises, adaptations, etc.). It is necessary (or at least advisable) for you to specify what exercises were performed and with what loads (dosage) or, at the very least, to indicate whether the exercise was high-impact, vigorous, or light, etc.
  2. There are errors in your bibliography:
    a) The number 1 appears twice as a citation, which disrupts the entire numbering sequence.
    b) References 6 and 11 are the same [Gomis M et al, 2009].

Author Response

Dear Reviewer,

Thank you for your valuable feedback. I have addressed each of your comments below.

  1. In my opinion, the MATERIALS and METHODS section is insufficient. You only mention one hour of exercise, but you do not specify the type of exercise performed, which joints were involved, or whether the exercise was high- or low-impact… and, obviously, you also do not provide any dosage of the exercise. You state that you followed the recommendations of Negrier et al. (2013), but these recommendations are very general (low-impact activities, safe exercises, adaptations, etc.). It is necessary (or at least advisable) for you to specify what exercises were performed and with what loads (dosage) or, at the very least, to indicate whether the exercise was high-impact, vigorous, or light, etc. As requested, we have provided a detailed description of the activities on page 4, lines 141-151, and page 5, lines 152-163. We believe that a more granular description of the exercises performed in each session would be excessive. These exercises were personalized for each participant according to their specific characteristics, target joints, and the progress observed throughout the study.
  2. There are errors in your bibliography: a) The number 1 appears twice as a citation, which disrupts the entire numbering sequence. b) References 6 and 11 are the same [Gomis M et al, 2009]. We appreciate you pointing out the error in the bibliographic numbering. The numbering has now been corrected throughout the manuscript

Reviewer 3 Report

Comments and Suggestions for Authors

The manuscript presents a well-designed pilot study investigating the safety and impact of controlled physical activity in individuals with severe hemophilia (PwH) under standard prophylaxis with a trough factor level of ≥20%. The study addresses an important gap in the literature by demonstrating that structured physical activity can be safely implemented without modifying prophylaxis regimens, but the study’s severe limitations—especially the tiny sample size, lack of controls, and high attrition—make it unsuitable for publication in its current form. The results are not robust enough to support the conclusions, and the methodological flaws undermine confidence in the findings.

  1. The study’s conclusions are based on only 9 participants, which severely limits statistical power and generalizability. Even as a pilot study, this sample is too small to draw meaningful conclusions about safety or efficacy.
  2. Without a control group (e.g., PwH not engaging in physical activity), it is impossible to determine whether observed changes (or lack thereof) are due to the intervention or natural variability.
  3. Participation dropped significantly after 6 months, with only 4 of 9 patients attending >60% of sessions in the latter half. This raises concerns about the feasibility of the intervention in real-world settings.
  4. The lack of statistical significance in most outcomes (e.g., HJHS, HEAD-US) contradicts the emphasis on "improvement trends," which are not robust enough to support clinical recommendations.
  5. The study design does not account for confounding variables (e.g., baseline joint damage severity, individual fitness levels).
  6. The findings do not significantly advance current knowledge. Previous studies (e.g., Negrier et al., 2013; Calatayud et al., 2020) have already established the safety of exercise in hemophilia, albeit with larger samples.
  7. Statistical analysis is underpowered—non-significant p-values dominate, yet the discussion emphasizes "trends" without caution.
  8. No protocol registration (e.g., ClinicalTrials.gov), raising concerns about post hoc analysis.

Author Response

Dear Reviewer,

Thank you for your comments. My point-by-point responses are below.

  1. The study’s conclusions are based on only 9 participants, which severely limits statistical power and generalizability. Even as a pilot study, this sample is too small to draw meaningful conclusions about safety or efficacy. We are conscious of the limited sample size; however, we believe our study provides valuable insights as a single-center pilot study involving patients with a rare disease. The participants shared highly comparable characteristics (all had severe hemophilia on regular prophylaxis with the same drug, a low ABR, and at least 50% preserved joint function) and were committed to a 12-month, twice-weekly sports activity program. We believe this makes our findings significant and a strong starting point for future research with a larger sample size. The safety observed in our participants supports the development of multi-center studies. In other words, our goal is not to provide general recommendations but to use these initial results as a foundation for developing future studies with greater statistical power.
  2. Without a control group (e.g., PwH not engaging in physical activity), it is impossible to determine whether observed changes (or lack thereof) are due to the intervention or natural variability. Each participant served as their own control. We were able to make this a within-subject design by collecting data on ABR and joint status (via HJHS and HEAD-US scores) at baseline, throughout the study, and at the final follow-up
  3. Participation dropped significantly after 6 months, with only 4 of 9 patients attending >60% of sessions in the latter half. This raises concerns about the feasibility of the intervention in real-world settings. While a higher adherence rate would certainly be appreciated in a clinical study, the observed decrease in participation is consistent with patterns seen in the general population. As suggested by the trainers who conducted the study, adherence is typically higher in the initial months of a new training program before gradually declining. Therefore, our data reflects real-world behavior and is not necessarily a limitation of the study design itself
  4. The lack of statistical significance in most outcomes (e.g., HJHS, HEAD-US) contradicts the emphasis on "improvement trends," which are not robust enough to support clinical recommendations. + Statistical analysis is underpowered—non-significant p-values dominate, yet the discussion emphasizes "trends" without caution. Once again, we discuss this as an observed trend. We underscored that these findings are not statistically significant and, consequently, should not be interpreted as a basis for general recommendations.
  5. The study design does not account for confounding variables (e.g., baseline joint damage severity, individual fitness levels). Table 2 presents the participants' baseline articular characteristics, including ABR, HJHS, and HEAD US scores. In contrast, Table 3 details the baseline results of the physical tests performed, which provide an indication of the participants' initial fitness level.
  6. The findings do not significantly advance current knowledge. Previous studies (e.g., Negrier et al., 2013; Calatayud et al., 2020) have already established the safety of exercise in hemophilia, albeit with larger samples. The study by Negrier et al., 2013 is a review that synthesizes evidence from several small studies to provide recommendations. Regarding the study by Calatayud et al., 2020, its focus was on improving strength and joint status over an 8-week period. The studies cited did not mention the factor levels used to perform physical activity, nor did it discuss an approach based on the study of PK, which is a key point of our research. We believe our study makes several novel contributions to the literature. First, its main objective was to assess safety, and the 12-month duration provides a more robust timeframe for evaluating both safety and changes in joint status. Second, our research specifically introduces the use of a 20% trough level as a safe threshold for physical activity, a detail not previously established in the literature. Finally, our approach is unique in that it does not alter the patient's standard prophylaxis schedule. Instead, it adapts the sports regimen to the individual's PK curve, integrating physical activity into their normal life without requiring changes to their treatment or without the need to infuse on the same day as the workout. We therefore believe our findings represent a significant advancement over what has been previously reported
  7. No protocol registration (e.g., ClinicalTrials.gov), raising concerns about post hoc analysis.

Round 2

Reviewer 2 Report

Comments and Suggestions for Authors

Although, in my opinion, the explanation of the exercise program and its dosage is probably still improvable, the revision and improvement of the paragraph where it is explained is appropriate and reasonable, considering this is a pilot study; therefore, I am satisfied with their response. In any case, for future, more extensive studies based on this pilot study—which I assume they will publish later on—I would recommend providing even more detail about this program, if possible, as it could be very valuable to identify protocols that can be safely replicated.

Author Response

Dear Reviewer

Thank you for your constructive comments and observations. My responses to your points are detailed below.

Comment 1:

Although, in my opinion, the explanation of the exercise program and its dosage is probably still improvable, the revision and improvement of the paragraph where it is explained is appropriate and reasonable, considering this is a pilot study; therefore, I am satisfied with their response. In any case, for future, more extensive studies based on this pilot study—which I assume they will publish later on—I would recommend providing even more detail about this program, if possible, as it could be very valuable to identify protocols that can be safely replicated.

Response:

As mentioned, this is a pilot study with several limitations. Future studies with larger sample sizes, longer durations and detailed training program will be necessary to more robustly assess the efficacy of physical activity in the management of hemophilia.

Reviewer 3 Report

Comments and Suggestions for Authors

While the authors provide thoughtful responses to the initial critique, the fundamental limitations of the study remain unresolved and significantly weaken its scientific validity and impact. Below are the keys, addressing the authors’ rebuttals:

  1.  While small samples are common in rare diseases, the lack of a control group, inadequate statistical power, and high attrition (only 4/9 participants maintained adherence) make it impossible to draw reliable conclusions—even as preliminary data.
  2. Even if participants shared similar characteristics, the absence of a comparator group means observed changes (e.g., stable ABR, HEAD-US trends) could reflect natural variability or placebo effects rather than the intervention.
  3. Without a parallel group of PwH not engaging in physical activity, the study cannot rule out confounding factors (e.g., regression to the mean, seasonal variations in joint health).
  4. A>50% attrition rate in a 12-month study severely limits interpretability. If most participants discontinued the intervention, the "safety" and "efficacy" claims are based on a non-representative subset. If the intervention cannot sustain participation even in a controlled trial, its clinical utility is doubtful.
  5. The manuscript repeatedly emphasizes "improvements" (e.g., HEAD-US score changes) despite non-significant p-values. This risks misleading readers into overinterpreting noisy data.
  6. The absence of a pre-registered protocol (e.g., on ClinicalTrials.gov) raises concerns about selective reporting of outcomes or analyses.

Author Response

Dear Reviewer

Thank you for your constructive comments and observations. My responses to your points are detailed below.

Comment 1:

 While small samples are common in rare diseases, the lack of a control group, inadequate statistical power, and high attrition (only 4/9 participants maintained adherence) make it impossible to draw reliable conclusions—even as preliminary data.”

Response:

We are conscious about study limitations but, according with definition of pilot study, this is a small-scale preliminary study aimed to assess the feasibility of the study's methods, procedures, and design, as well as to identify potential problems, as suggested by your comments. Future studies should provide controlled designs, larger sample sizes, improved adherence and longer durations in order to draw reliable conclusions.

Comment 2:

Even if participants shared similar characteristics, the absence of a comparator group means observed changes (e.g., stable ABR, HEAD-US trends) could reflect natural variability or placebo effects rather than the intervention.”

Response:

We fully acknowledge that the absence of a control group represents a methodological limitation of this pilot study. However, the primary objective was to assess the safety and feasibility of a structured physical activity program in people with severe hemophilia (PwH) already on standard prophylaxis, without any modifications to their treatment regimen. To minimize internal variability, we selected a homogeneous sample in terms of clinical and joint status characteristics. In this context, the stability of the ABR and joint health scores, along with the absence of bleeding events, support the safety of the intervention. That said, we agree that without a comparator group, we cannot rule out that some of the observed changes may be due to natural fluctuations or placebo effects. This limitation is now explicitly addressed in the revised manuscript under the section "Limitations of the Study".

Comment 3:

Without a parallel group of PwH not engaging in physical activity, the study cannot rule out confounding factors (e.g., regression to the mean, seasonal variations in joint health).”

Response:

We agree that, in the absence of a parallel group of PwH not participating in physical activity, it is not possible to entirely exclude the influence of confounding factors, such as spontaneous changes in joint health over time or potential seasonal variation. Nonetheless, the structured follow-up at three time points (baseline, 6 months, and 12 months), combined with the personalized scheduling of physical activity based on each patient’s pharmacokinetic profile, allowed for a consistent and clinically controlled evaluation. Notably, a reduction in synovitis was observed in six out of nine patients by the six-month timepoint, which may plausibly be attributed to the intervention. This improvement was not sustained in the second half of the study, coinciding with a decline in adherence to the training program, further supporting a possible link between physical activity and clinical benefit. This limitation and its implications are also clearly stated in the revised manuscript under the “Limitations of the Study” section, along with a call for future randomized controlled studies.

Comment 4:

A>50% attrition rate in a 12-month study severely limits interpretability. If most participants discontinued the intervention, the "safety" and "efficacy" claims are based on a non-representative subset. If the intervention cannot sustain participation even in a controlled trial, its clinical utility is doubtful.”

Response:

We agree that decline in participation represents a limit for the findings. Neverthless, attrition rate is similar to the general population's involvement in recreational sports at the Sisport Sports Center, underlying ubiquitous approach to physical activity in the population. For further comments on this point please refers to previous respose.

Comment 5:

The manuscript repeatedly emphasizes "improvements" (e.g., HEAD-US score changes) despite non-significant p-values. This risks misleading readers into overinterpreting noisy data”

We underlined the limitation of a small number of partecipants at the first begininng of the discussion, we expose our findings as a trend that cannot be endorsed by significant p-values due to small sample size.

Comment 6:

The absence of a pre-registered protocol (e.g., on ClinicalTrials.gov) raises concerns about selective reporting of outcomes or analyses.”

The study was submitted to the Local Ethics Commitee (Città della Salute e della Scienza, University Hospital of Turin Territorial Ethics Commitee) on June 25, 2019 and received the approval on October 10, 2019 (protocol number 0098190). Similar Ethics Commitee approvals are reported for other studies previously published in JCM. The study was conducted according to Good Clinical Practice, furthermore data were analyzed separately by haematologists and trainers and they were matched at the end of the study period.